# Exploration by Learning Diverse Skills through Successor State Representations

**Paul-Antoine Le Tolguenec**
ISAE-Supaero, Airbus
`paul-antoine.le-tolguenec@airbus.com`

**Yann Besse**
Airbus
`yann.besse@airbus.com`

**Florent Teichteil-Konigsbuch**
Airbus
`florent.teichteil-konigsbuch@airbus.com`

**Dennis G. Wilson**
ISAE-Supaero, Université de Toulouse
`dennis.wilson@isae-supaero.fr`

**Emmanuel Rachelson**
ISAE-Supaero, Université de Toulouse
`emmanuel.rachelson@isae-supaero.fr`

## Abstract

The ability to perform different skills can encourage agents to explore. In this work, we aim to construct a set of diverse skills that uniformly cover the state space. We propose a formalization of this search for diverse skills, building on a previous definition based on the mutual information between states and skills. We consider the distribution of states reached by a policy conditioned on each skill and leverage the successor state representation to maximize the difference between these skill distributions. We call this approach LEADS: Learning Diverse Skills through Successor State Representations. We demonstrate our approach on a set of maze navigation and robotic control tasks which show that our method is capable of constructing a diverse set of skills which exhaustively cover the state space without relying on reward or exploration bonuses. Our findings demonstrate that this new formalization promotes more robust and efficient exploration by combining mutual information maximization and exploration bonuses.

## 1 Introduction

Humans demonstrate an outstanding ability to elaborate a repertoire of varied skills and behaviors, without extrinsic motivation and supervision. This ability is currently captured through the problem of unsupervised skill discovery in reinforcement learning [41], where one seeks to learn a finite set of policies in a given environment whose behaviors are notably different from each other. Implicitly, seeking behavioral diversity is also related to the question of efficient exploration, as a set of skills that better covers the state space is often preferable. Seeking diversity has recently been successfully studied through the prism of mutual information maximization [17, 13, 8]. In this article, we argue maximizing mutual information might be ambiguous when seeking exploratory behaviors and propose an alternative, motivated variant. The key intuition underpinning this formulation is that a good set of exploratory skills should maximize state coverage, while preserving the ability to distinguish a skill from another. Then, we propose a new algorithm which implements this generalized objective, leveraging neural networks as estimators of the state occupancy measure. This algorithm demonstrates better exploration properties than state of the art methods, both those designed for reward-free exploration and those seeking skill diversity.

38th Conference on Neural Information Processing Systems (NeurIPS 2024).

We motivate our developments with a first illustrative toy example. Let us consider a reward-free Markov decision process [38, MDP] $(\mathcal{S}, \mathcal{A}, P, \gamma, \delta_0)$ where $\mathcal{S}$ is the state space, $\mathcal{A}$ the action space, $P$ the transition function, $\gamma$ the discount factor and $\delta_0$ the initial state distribution. A behavior policy, parameterized by $\theta$, maps state to distribution over actions. We define a *skill encoding* $z \in \mathbb{R}^d$ as an abstract set of skill descriptors that enhances the policy description and conditions the action taken $a \sim \pi_\theta(s, z)$, making the policy a function $\pi : \Theta \times \mathcal{S} \times \mathbb{R}^d \mapsto \Delta(\mathcal{A})$. Unsupervised skill discovery generally considers a finite collection $\mathcal{Z} = \{z_i\}_{i \in [1,n]}$ of $n$ skills, and a distribution $p(z)$ on skills within $\mathcal{Z}$ (generally uniform). For a given $\theta$, each skill $z$ induces a state distribution $p(s|z)$. The state distribution under the full set of skills is hence $p(s) = \sum_{\mathcal{Z}} p(s|z)p(z)$. When maximizing diversity, we aim to find $\theta^*$ such that, for any pair of skills $z_1, z_2 \in \mathcal{Z}$, the states visited by each skill are as separable as possible. Additionally, we would like the state coverage of the full set of skills to cover as much of the state space as possible.

Maximizing diversity has been formalized in the literature as the maximization of the mutual information $\mathcal{I}(S, Z)$ between the state random variable $S$ and the skill descriptor $Z$. This mutual information [39, MI] quantifies how predictable $S$ is, when $Z$ is known (and conversely):

$$\mathcal{I}(S, Z) \triangleq D_{KL}(\mathbb{P}(S, Z) || \mathbb{P}(S)\mathbb{P}(Z)),$$
$$= \mathcal{H}(S) - \mathcal{H}(S|Z),$$

where $D_{KL}$ is the Kullback-Leibler divergence and $\mathcal{H}$ is the (conditional) entropy.

Figure 1 illustrates why $\mathcal{I}(S, Z)$ is an imperfect measure of diversity. In this figure, two sets $\mathcal{Z}_1$ and $\mathcal{Z}_2$ of four skills each are represented by their respective state coverage in a grid maze

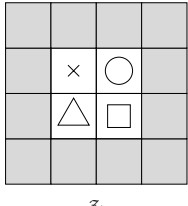 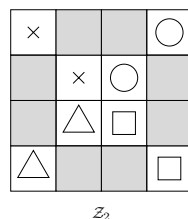

Figure 1: State distributions of two sets $\mathcal{Z}_1$ (left) and $\mathcal{Z}_2$ (right) of four skills each on a grid maze. Each skill's visited states are represented by a different symbol and distributed uniformly. The gray boxes are unreachable.

MDP (one symbol per skill). In each set of skills, each skill is picked with probability $p(z) = 1/4$. On the leftmost image ($\mathcal{Z}_1$), each skill covers a single state, hence $p(s|z) = 1$ in this state and 0 otherwise. On the rightmost image ($\mathcal{Z}_2$), $p(s|z) = 1/2$ on each covered state and 0 otherwise. Consequently, $\mathcal{I}_1(S, Z) = \mathcal{H}_1(S) - \mathcal{H}_1(S|Z) = \log 4$ and $\mathcal{I}_2(S, Z) = \mathcal{H}_2(S) - \mathcal{H}_2(S|Z) = \log 4$. The mutual information is hence ambiguous as it ranks the two sets of skills equally, while $\mathcal{Z}_2$ enables better exploration.

Our contribution is structured around maximizing a variant of $\mathcal{I}(S, Z)$, using the successor state representation [4] to enable its estimation. Section 2 puts this idea in perspective of the relevant literature, highlighting similarities and contrasts with previous works. Then Section 3 introduces and discusses our LEADS algorithm. Section 4 reports empirical evidence of the good exploratory properties of LEADS, demonstrating its ability to outperform state-of-the-art methods on a span of benchmarks.

## 2    Related work

This section endeavors to provide the reader with a comprehensive overview of approaches which aim at exploration in RL and skill diversity.

**Exploration bonuses.** Driving exploration by defining reward bonuses has been a major field of research for over two decades [25, 6, 1, 3, 36, 7, 2, 19]. These methods generally involve defining reward bonuses across the state space, which decrease over time in well-explored states. In large state spaces where count-based methods are challenging, these approaches require function approximation to generalize across the state space. However, as explained by Jarrett et al. [23], identifying these exploration bonus approximators is an adversarial optimization problem making these approaches unstable in practice [12].

**Quality Diversity.** Exploration can also be achieved as a by-product of behavior diversity search. Evolutionary algorithms have been used to search for new behaviors, such as in Novelty Search [27], and for a diversity of behaviors which encourage high performance through the growing domain

of Quality Diversity algorithms, exemplified by MAP-Elites [11]. In these algorithms, skills are usually characterized through hand-crafted *behavior descriptors*, although there are methods which learn such descriptors through variational inference [10, 9]. Diversity is expressed as the coverage over such descriptors, and policies are modified through gradient-free [11] or gradient-based [32] updates in order to incrementally populate an archive of diverse skills. While these algorithms have demonstrated competitive results with exploration algorithms in RL [9], in general, they require expertise to define a behavior descriptor and are less sample-efficient than RL methods [28].

**Information theory approaches to diversity.** The field of information theory offers valuable insights into designing algorithms that generically promote diversity. Gregor et al. [17], Eysenbach et al. [13] and Sharma et al. [40] were pioneers in proposing methods to maximize either the reverse or forward form of $\mathcal{I}(S, Z)$ for a fixed set of skills $\mathcal{Z}$. To address the issue of unbalanced state distributions across skills throughout the entire environment, Campos et al. [8] suggest sequencing the problem into three stages: pure exploration in $\mathcal{S}$ to promote sampled states diversity, state clustering via variational inference, and skill learning by maximisation of the forward form as proposed by Sharma et al. [40]. In the exploration phase, Campos et al. [8] employs the State Marginal Matching algorithm proposed by Lee et al. [26], which resembles MI-based algorithms as it combines mutual information maximization with an exploration bonus to efficiently explore the state space.

Focusing even more on exploration, Liu and Abbeel [29] automate learning Novelty Search behavior descriptors by training an encoder with contrastive learning in order to promote exploration. Focusing also on exploration, Kamienny et al. [24] employ an incremental approach combined with the maximization of the reverse form of MI to eliminate skills that are not discernable enough, subsequently generating new skills from the most discernable ones.

Finally, recent studies including those by Park et al. [33], Park et al. [34], and Park et al. [35], integrate mutual information (MI) with trajectory-based metrics between states to enhance exploration. Specifically, these studies aim to maximize the Wasserstein distance, also known as the Earth mover's distance, between $\mathbb{P}(S, Z)$ and $\mathbb{P}(S)\mathbb{P}(Z)$ while the definition of MI uses the KL divergence. This distance requires specifying a metric for the state space. Park et al. [33] and Park et al. [34] uses the Euclidean distance, while Park et al. [35] employs the temporal distance.

It is important to note that, although these methods induce an interesting approach to exploration (as a by-product of diversity), they do not explicitly aim at encouraging coverage of the MDP's state space.

**Successor features for MI maximization.** Few attempts have been made to exploit universal successor features [5] in MI maximization methods. Machado et al. [30] decompose the learning process into two stages: first, they automatically acquire specific options, which are then used to develop a hierarchical policy that integrates these options. In contrast, Hansen et al. [21] first learn a general task-based policy conditioned on a task $z$, and subsequently optimize the reverse form of MI by utilizing universal successor features to identify a range of diverse skills efficiently. More recently, Grillotti et al. [18] used successor features to optimize the quality of predefined behaviors based on the Quality Diversity formulation, which requires hand-crafted behavior descriptors.

The method we propose explicitly aims at promoting diversity with the goal of achieving good exploration. For this purpose, we turn the MI objective function into a new objective which promotes exploration, separating this work from previous contributions in MI maximization.

## 3    Promoting exploration with successor state representations

We now derive an algorithm that explicitly encourages exploration in diversity search. We start by casting mutual information maximization in terms of state occupancy measures and derive a lower bound which can be maximized with stochastic gradient ascent on the policy's parameters $\theta$ (Section 3.1). Then, as maximizing MI does not encourage exploration, we transform this lower bound so that its maximization fits this goal, leading to a new target quantity designed to promote both diversity and exploration (Section 3.2). This enables defining and implementing a general algorithm for *Learning Diverse Skills through successor state representations* (LEADS, Section 3.3).

## 3.1 MI lower bound with successor state representations

The mutual information $\mathcal{I}(S, Z)$ is naturally expressed through the (conditional) probability densities of $S$ and $Z$. Successor state representation estimators [4, SSR] provide a direct way of estimating these densities. More precisely, we identify the Successor State Representation as the density of the measure defined on the set of states. The SSR $p(s|s_1, \theta, z)$ of a skill $(\theta, z)$ is the state occupancy measure of policy $\pi_\theta(s, z)$, when starting from $s_1$, that is

$$p(s_2|s_1, \theta, z) = \sum_{t=0}^{\infty} \gamma^t p \left( s_t = s_2 \;\middle|\; \begin{array}{l} s_0 = s_1, \\ a_t \sim \pi_\theta(s_t, z) \end{array} \right). \tag{1}$$

For the sake of readability, we make $\theta$ implicit in our notations going forward. This measure can be understood as the probability density of reaching the state $s_2$ starting from another state $s_1$ when the skill $z$ is used [22]. It encodes all the paths of the Markov chain that can be taken from $s_1$ to $s_2$ when the skill $z$ is used [4]. Here $(s_1, s_2) \in \mathcal{S}^2$ can be any two states of the Markov chain.

The SSR is the fixed point of a Bellman operator and one can employ temporal difference (TD) learning to guide a function approximator towards this fixed point. There are numerous methods for estimating the SSR density [4, 22, 14, 15]. In this paper, we use C-Learning [14] for estimating the SSR for all experiments. C-Learning proposes to cast the regression problem of learning $p$ as a classification problem. Specifically, for a given pair $(s_1, a, z)$, a classifier $\sigma(f_\phi(s_1, a, s_2, z))$, where $\sigma$ is the sigmoid function, is trained to predict whether $s_2$ was sampled from an outcome trajectory from $s_1$ ($s_2 \sim p(\cdot|s_1, a, z)$) or from the marginal density of all possible skills ($s_2 \sim p(s)$, the distribution over $\mathcal{S}$). The optimal classifier holds the following relation with the SSR's density: $e^{f_\phi(s_1, a, s_2, z)} = p(s_2|s_1, a, z)/p(s_2)$. We denote $m_\phi(s_1, a, s_2, z) = e^{f_\phi(s_1, a, s_2, z)}$, as the SSR estimated via C-Learning, taking $a \sim \pi_\theta(s_1, z)$. For the sake of brevity, we will use the notation $m_\phi(s_1, s_2, z) = \mathbb{E}_{a \sim \pi_\theta(s_1, z)} [m_\phi(s_1, a, s_2, z)]$. We note that any method which reliably estimates $m_\phi(s_1, s_2, z)$ can be used instead of C-Learning in LEADS.

As described in Section 1, the mutual information between the state random variable $S$ and the skill descriptor $Z$ is $\mathcal{I}(S, Z) = D_{KL}(\mathbb{P}(S, Z) || \mathbb{P}(S)\mathbb{P}(Z))$. This can be expressed as

$$\mathcal{I}(S, Z) = \mathbb{E}_{(s_2, z) \sim p(s, z)} \left[ \log \left( p(s_2|z)/p(s_2) \right) \right]. \tag{2}$$

State $s_2$ can be any state $s$ within the set $\mathcal{S}$, yet we refer to it as $s_2$ to ensure that when we incorporate the definition of the SSR, it aligns directly with the notation of Equation 1. By definition of the SSR under a given policy $\pi(s, z)$, the state distribution $p(s|z)$ obeys

$$p(s_2|z) = \mathbb{E}_{s_1 \sim p(s|z)} \left[ p(s_2|s_1, z) \right]. \tag{3}$$

By sampling $s_1 \sim p(s|z)$, we can therefore estimate $p(s_2|z)$, allowing for the injection of $m(s_1, s_2, z) = p(s_2|s_1, z)/p(s_2)$ from Eysenbach et al. [14] into Equation 2:

$$\mathcal{I}(S, Z) = \mathbb{E}_{(s_2, z) \sim p(s, z)} \left[ \log \left( \mathbb{E}_{s_1 \sim p(s|z)} \left( m(s_1, s_2, z) \right) \right) \right].$$

A Monte Carlo estimator of $\mathcal{I}(S, Z)$ can be obtained by sampling $(s, z)$ from a replay buffer. However, for every state $s$ sampled this way, the estimate of $p(s|z)$ provided by Equation 3 requires sampling a batch of $s_1 \sim p(s|z)$. This is possible through keeping separate replay buffers for each skill, but would be computationally intractable for an accurate estimation. Hence, as in similar methods, we turn rather to a lower bound on $\mathcal{I}(S, Z)$, which admits a Monte Carlo estimator, and which will be maximized with respect to $\theta$ using stochastic gradient ascent.

Applying Jensen's inequality to the inner expectation in the previous expression yields

$$\mathcal{I}(S, Z) \geq \mathbb{E}_{\substack{z \sim p(z) \\ s_2 \sim p(s|z) \\ s_1 \sim p(s|z)}} \left[ \log(m(s_1, s_2, z)) \right]. \tag{4}$$

Note that $\theta$ participates in this lower bound through $m(s_1, s_2, z) = \mathbb{E}_{a \sim \pi_\theta(\cdot|s_1)} m(s_1, a, s_2, z)$. This quantity is essentially what an algorithm like DIAYN [13] maximizes: given separate experience buffers for each skill, one can compute a Monte Carlo estimate of this lower bound on the gradient of the mutual information $\mathcal{I}(S, Z)$ with respect to $\theta$.

## 3.2 Promoting exploration with different skills

Looking closely at Equation 4, one can note that the interplay between each skill's state distribution is lost when deriving the lower bound. Hence, this bound can be maximized without actually pushing skills towards distinct states, hence without skill diversity. To regain the incentive to promote skill diversity, we propose to encourage exploration in each skill to put probability mass on states which receive little coverage by the full set of skills. In other words, for a given transition $(s, \pi(s))$, we want to augment the probability of occurrence for one skill while decreasing it for all others.

Since $m(s_1, s, z) \geq 0$, we can introduce the following bound:

$$\mathcal{I}(S, Z) \geq \mathop{\mathbb{E}}_{\substack{z \sim p(z) \\ s_2 \sim p(s|z) \\ s_1 \sim p(s|z)}} \left[ \log \left( \frac{m(s_1, s_2, z)}{1 + \sum\limits_{z' \in \mathcal{Z}} m(s_1, s_2, z')} \right) \right]. \tag{5}$$

While Equation 5 is a looser lower bound than Equation 4, it goes towards the goal of diversity search originally captured by MI maximization: to have distinct state distributions for each skill.

However, as illustrated in the introductory example, maximizing MI does not promote large state coverage, nor exploratory behaviors. Equation 5 promotes skill diversity, but it does not explicitly encourage state coverage. We therefore argue that exploratory behaviors can be obtained by focusing the state distribution of each skill towards specific states within the support of $p(s|z)$. Instead of sampling $s$ according to $p(s|z)$, we encourage exploration by attributing more probability mass to states that trigger exploration. We do so by creating a sampling distribution $\delta(s|z)$ based on an uncertainty measure $u_t(s, z)$, as explained next.

Biasing a skill towards promising states for exploration is often achieved through exploratory bonuses based on uncertainty measures [36, 7, 2], or repulsion mechanisms [16]. These heuristic measures encourage exploration by pushing policies towards states of high uncertainty or away from previously covered transitions. The novelty of our approach is the use of an exploration bonus in the framing of mutual information maximization. However, by replacing $s_2 \sim p(s|z)$ by another sampling distribution $\delta(s|z)$, we lose the theoretic guarantee of maximizing the lower bound of Equation 5. Rather, we propose a new objective, expressing an incentive to explore within the search for skill diversity, but with a greater focus on state coverage through distinct skills than previous lower bounds on mutual information. The quantity we maximize is then:

$$\mathcal{G}(\theta) = \mathop{\mathbb{E}}_{\substack{z \sim p(z) \\ s_1 \sim p(s|z) \\ a_z \sim \pi_\theta(\cdot|s_1, z) \\ s_2 \sim \delta(s|z)}} \left[ \log \left( \frac{m(s_1, a_z, s_2, z)}{1 + \sum\limits_{z' \in \mathcal{Z}} m(s_1, a_{z'}, s_2, z')} \right) \right]. \tag{6}$$

The uncertainty measure $u_t$ which defines the distribution $\delta(s|z)$ is designed to explore under-visited areas and to create repulsion between different skills. We define three desired properties for states to prioritize using this measure. To describe these properties, consider a sequence of policies $\pi_t$, each defining a set of skills $\{\pi_t(\cdot, z)\}_{z \in \mathcal{Z}}$, and the corresponding sequence of SSRs $m_t$; the three properties are then as follows. (1) A good target state $s_t^z$ for skill $z$ at time step $t$ is one that has high probability of being visited by $\pi_t(\cdot, z)$, but was relatively infrequently visited by previous policies' state occupancy measures $\{m_k\}_{k \in [1, t-1]}$ for any skill. (2) It is also a state which has both high probability of being reached by the current $\pi_t(\cdot, z)$ and low probability of being reached by any other current skill, starting from $s_0$. (3) Given a previous target state $s_{t-1}^z$, a good new target state is one that has high (resp. low) probability to be reached by $\pi_t(\cdot, z)$ (resp. any other skill), starting from $s_{t-1}^z$. While the first property explicitly encourages visiting under-visited states, the two others do not encourage exploration per se, and rather strengthen skill diversity by pushing their state distributions apart. This leads to the idea of ranking target states according to:

$$u_t(s, z) = \underbrace{\log \left( \frac{m_t(s_0, s, z_i)}{\sum_{k=1}^{t-1} \sum_{z'} m_k(s_0, s, z')} \right)}_{\text{Explore under-visited areas}} + \underbrace{\sum_{z' \neq z} \log \left( \frac{m_t(s_{t-1}^z, s, z)}{m_t(s_{t-1}^{z'}, s, z')} \right) + \log \left( \frac{m_t(s_0, s, z)}{m_t(s_0, s, z')} \right)}_{\text{Repulsion between skills}}$$

$$\tag{7}$$

The $\delta(s|z)$ distribution of Equation 6 therefore allocates more probability to states with high $u_t(s, z)$. Empirically, we found that making $\delta$ deterministic as the Dirac distribution on a state that maximizes $u_t(s, z)$ enabled efficient exploration. Appendix A provides a more formal perspective on the derivation of the $u_t$ uncertainty measure.

## 3.3 The LEADS algorithm

With the objective of maximizing $\mathcal{G}(\theta)$ (Equation 6), we define the LEADS (Learning Diverse Skills through successor state representations) algorithm, presented in Algorithm 1.

In order to maximize $\mathcal{G}(\theta)$, we must first sample from $p(s|z)$ for each skill $z$, then compute the SSR for the current skills $\pi_t(\cdot, z)$ in order to define $\mathcal{G}(\theta)$, then finally update $\theta$ to maximize $\mathcal{G}(\theta)$. As such, an iteration of LEADS features three phases in order to learn a new set of skills: sampling, learning the SSR, and optimizing the policy parameters $\theta$.

First, the sampling phase populates separate replay buffers $\mathcal{D}_z$ for each skill $z \sim p(z)$. This is achieved by rolling out $n_{ep}$ episodes with a given policy $\pi_\theta(\cdot, z)$. Then, C-learning is used to compute the parameters $\phi_t$ of a joint SSR $m_t(s_1, a, s, z)$ for $\pi_t$, and store it. We note that the on-policy version of C-learning can be used for this step as the data in the replay buffers has been collected with the current policy $\pi_t$. This SSR, and all previous ones, are then used to define $u_t(s, z)$, which in turn defines the distribution

---

**Algorithm 1** LEADS

Initialize $\theta_0$
**for** $t \in [0, N]$ **do**
    # Collect samples
    $\mathcal{D}_z = \emptyset, \forall z \in \mathbb{Z}$
    **for** $e \in [1, n_{\text{ep}}]$ **do**
        Sample skill $z \sim p(z)$
        $\{(s_t, a_t, r_t, s'_t)\}$ = episode with $\pi_{\theta_t}(\cdot, z)$ from $s_0$
        $\mathcal{D}_z = \mathcal{D}_z \cup \{(s_t, a_t, r_t, s'_t)\}$
    **end for**
    # Learn the SSR
    Learn $m_{\phi_t}$ for $\pi_{\theta_t}$ using on-policy C-learning
    Sample $s \sim \delta(s|z)$
    # Improve $\theta$
    **for** $i \in [1, n_{\text{SGD}}]$ **do**
        Sample $z \sim p(z), s_1 \sim p(s|z)$
        $\theta \leftarrow \theta + \alpha \nabla_\theta [\mathcal{G}(\theta) + \lambda_h \mathcal{H}(\theta)]$
        Update $\phi_t$ using off-policy C-learning
    **end for**
**end for**

---

$\delta(s, z)$. As this can be approximated for a static policy, sampling can be performed before updating $\theta$. Sampling $s \sim \delta(s|z)$ is performed by running all states in $\mathcal{D}_z$ through $u_t(s, z)$, although, to limit computational cost, we can restrict this evaluation and selection to only a uniformly drawn subset of each $\mathcal{D}_z$.

We therefore have the necessary components to optimize $\theta$ according to $\mathcal{G}(\theta)$: a means of sampling states and the SSR. At each gradient ascent step, we sample a mini-batch of states $s_1$ from $\mathcal{D}_z$ for a given skill $z \sim p(z)$ to estimate $p(s|z)$. This permits calculating the objective $\mathcal{G}(\theta)$ for the current $\theta$ using $m_{\phi_t}$ and performing a gradient step. In the gradient calculation, we also include an action entropy maximization term, as done in other works [31, 20]:

$$\mathcal{H}(\theta) = \mathop{\mathbb{E}}_{\substack{s_1 \sim p(s|z) \\ z \sim p(z) \\ a \sim \pi(s_1, z)}} [-\log(\pi_\theta(a|s_1, z))]. \tag{8}$$

$\theta$ is therefore updated to maximize $\mathcal{G}(\theta) + \lambda_h \mathcal{H}(\theta)$, although $\lambda_h$ is intentionally kept small (0.05) to focus on the principal LEADS objective $\mathcal{G}(\theta)$.

Finally, after each gradient step, the SSR is updated using the off-policy formulation of C-learning so that $m_{\phi_t}$ remains representative of the state distribution under $\pi_{\theta_t}$. This is done without sampling new transitions and is justified by the fact that the target state $s_t^z$ and the states along a trajectory to $s_t^z$ are already within the replay buffer $\mathcal{D}_z$. $n_{SGD}$ steps of gradient ascent on $\theta$ are performed in this way, before a new iteration of LEADS is started.

# 4 Experiments & Results

We demonstrate LEADS on a set of maze navigation and robotic control tasks and compare its behavior to state-of-the-art exploration algorithms. We provide all code for LEADS and the baseline algorithms, as well as the scripts to reproduce the experiments (repository). All hyperparameters are summarized in Appendix C.

## 4.1 Evaluation benchmarks

**Mazes.** Maze navigation has been frequently used in the exploration literature, as 2D environments allows for a clear visualization of the behaviors induced by an algorithm. The assessment of state coverage is also easier to understand than in environments with high-dimensional states. We design three different maze navigation tasks, named **Easy**, **U**, and **Hard** (depicted in Figure 2), of increasing difficulty in reaching all parts of the state space. In each of the mazes, the state space is defined by the agent's Euclidean coordinates $\mathcal{S} = [-1, 1]^2$ and the action space corresponds to the agent's velocity $\mathcal{A} = [-1, 1]^2$. Hitting a wall terminates an episode, making exploration difficult.

**Robotic control.** We further assess the capabilities of LEADS in complex robotic control tasks. These tasks allow evaluating the ability of LEADS to explore in diverse and high-dimensional state spaces. We evaluate LEADS on a variety of MuJoCo [42] environments from different benchmark suites. **Fetch-Reach** [37] is a 7-DoF (degrees of freedom) robotic arm equipped with a two-fingered parallel gripper; its observation space is 10-dimensional. **Fetch-Slide** extends the former with a puck placed on a platform in front of the arm, increasing the observation space dimension to 25. **Hand** [37] is a 24-DoF anthropomorphic robotic hand, with a 63-dimensional observation space. **Finger** [44] a 3-DoF, 12-dimensional observation space, manipulation environment where a planar finger is required to rotate an object on an unactuated hinge. Appendix D discusses additional experiments and limitations on other MuJoCo tasks.

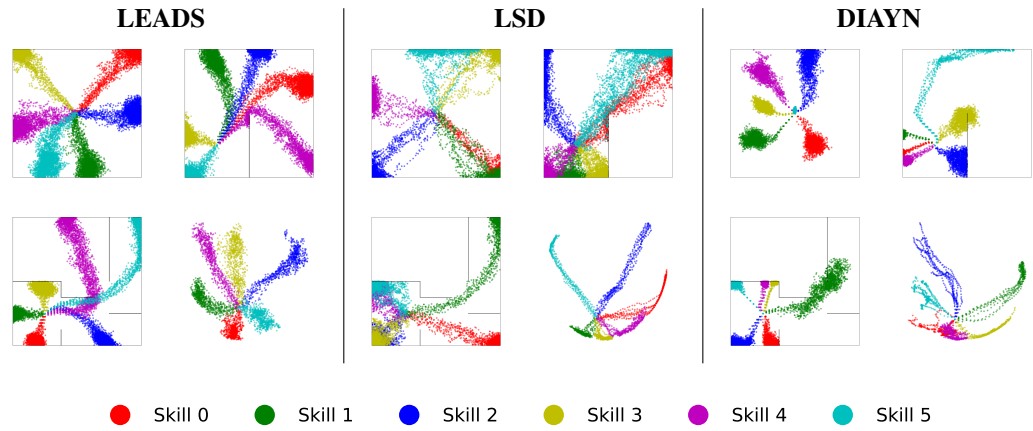

Figure 2: Skill visualisation for each algorithm. Per algorithm, the tasks are the mazes **Easy** (top left), **U** (top right), **Hard** (bottom left), and the control task **Fetch-Reach** (bottom right).

## 4.2 Skill Visualization

The first goal of this work, as exposed in the introductory example, is to obtain a set of skills that are as distinct from each other as possible in their visited states, while simultaneously covering as much of the state space as possible. We first assess this property in LEADS, DIAYN [13] and LSD [33] through a visual analysis of their state space coverage in Figure 2. For all experiments, the number of skills is $n_{\text{skill}} = 6$. To ensure fairness, for each algorithm, we report the skill visualization from the experiment that achieves maximum coverage (as defined in the following section) out of five runs. For each algorithm, Figure 2 presents four figures: the three mazes and the Fetch-Reach environment. In all the mazes, we visualize the 2D coordinates of the states reached over training. Given that 2D visualization is not suitable for the Fetch-Reach environment due to its higher state space dimension,

we project the state onto the two first eigenvectors of a Principal Component Analysis (PCA) of the states encountered by all skills in this environment.

We note that LEADS clearly defines distinct skills in the state space. Furthermore, it leads to a more extensive exploration of the environment than LSD and DIAYN. The Hard maze (bottom left) is noteworthy, as some parts of the environment are difficult to access due to bottleneck passages in the maze. LEADS is the only algorithm that manages to reach all sections of the Hard maze.

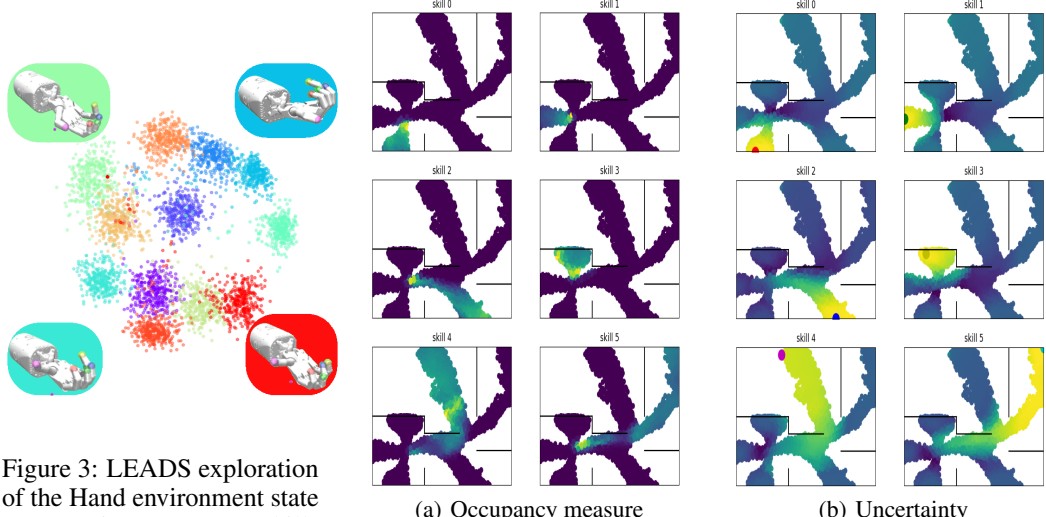

Figure 3: LEADS exploration of the Hand environment state space, using $n_{\text{skill}} = 12$ skills and a PCA over all explored states.

(a) Occupancy measure        (b) Uncertainty

Figure 4: (a): The SSR $m(s_0, s, z)$ at a given epoch during training on Hard maze, per skill, normalized in $[0, 1]$. (b): The uncertainty measure $u(s, z)$ at the final epoch on Hard maze, per skill, with the maximum state highlighted.

Figure 3 further demonstrates the ability of LEADS to create distinct skills in the high-dimensional state space of the Hand environment. It reports the state distribution obtained by $n_{\text{skill}} = 12$ skills found by LEADS, projected onto the two first components of a PCA on all visited states. As in the maze navigation tasks, LEADS explores states in distinct clusters, organized by skill. These skills correspond to a visual variety of finger positions, illustrating the ability to discover varied behaviors.

Section 3.2 claimed that the uncertainty measure $u_t(s, z)$ of Equation 7, and its associated distribution $\delta$ triggered relevant exploratory behaviors. Figures 4(a) and 4(b), display the final SSR and uncertainty measure, respectively, for the Hard maze task. The same visual representation of these densities for the other environments can be found in Appendix B. Figure 4(a) demonstrates that the SSR clearly separates states based on skill. The SSR is zero for all states outside of the distribution of a given skill, and all states in a skill have a non-zero probability of being reached by that skill. This confirms that this SSR can reliably be used within the objective function of Equation 6.

Similarly, Figure 4(b) illustrates how the uncertainty measure concentrates on states that promote both exploration and diversity. High values for this measure are found in very different states for each skill (hence promoting diversity), and within a skill, the highest value is found far from the starting state $s_0$ (hence promoting exploration). The state maximizing $u_t$ is represented by a colored dot for each skill. This confirms that the uncertainty measure achieves the joint goal of exploring under-visited areas and creating repulsion between skills. By defining $\delta(s|z)$ as the state which maximizes $u_t(s, z)$ for each skill $z$, LEADS ensures a continuing exploration of the state space.

### 4.3 Quantitative Evaluation

We now turn to a quantitative evaluation of the ability to explore using diversified skills. For the sake of completeness, we compare LEADS to seven seminal algorithms. Five of these are skill-based, namely DIAYN [13], SMM [26], LSD [33], CSD [34], and METRA [35]. We use $n_{\text{skill}} = 6$ for each skill-based algorithm and for all environments. The two last algorithms are pure-exploration

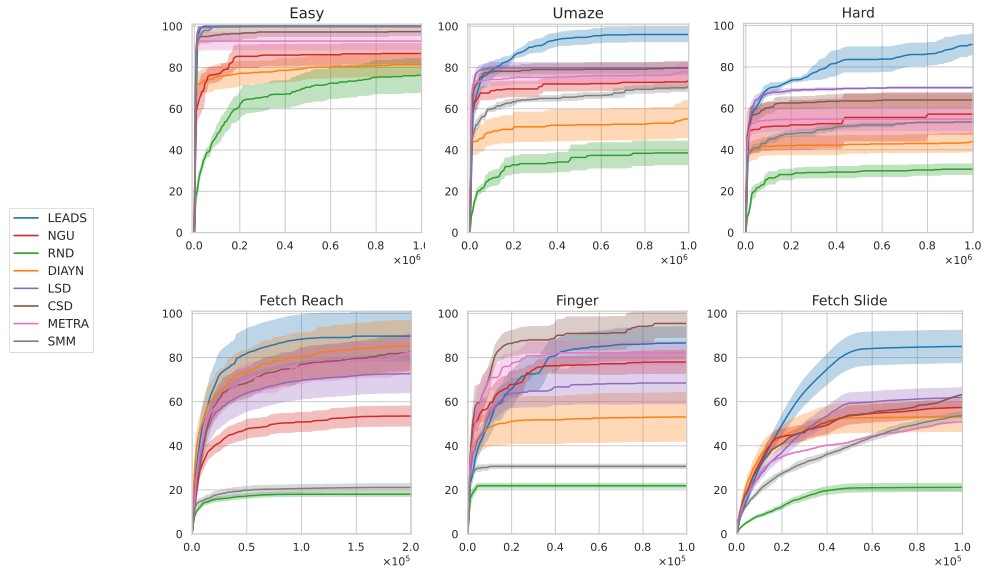

Figure 5: Relative coverage evolution across six tasks. The x-axis represents the number of samples collected since the algorithm began.

| Method | Easy (%) | Umaze (%) | Hard (%) | Fetch Reach (%) | Finger (%) | Fetch Slide (%) |
|--------|----------|-----------|----------|-----------------|------------|-----------------|
| RND | $76.6 \pm 7.3$ | $39.6 \pm 5.4$ | $30.8 \pm 4.3$ | $17.6 \pm 2.7$ | $21.3 \pm 1.6$ | $21.6 \pm 3.2$ |
| DIAYN | $81.4 \pm 8.6$ | $55.0 \pm 8.2$ | $43.8 \pm 4.5$ | $85.6 \pm 8.7$ | $53.8 \pm 12.3$ | $52.3 \pm 8.5$ |
| SMM | $100.0 \pm 0.0$ | $70.6 \pm 3.7$ | $53.4 \pm 0.5$ | $22.3 \pm 5.5$ | $31.2 \pm 1.4$ | $53.3 \pm 3.4$ |
| NGU | $86.8 \pm 8.4$ | $73.4 \pm 6.0$ | $57.2 \pm 8.3$ | $53.4 \pm 4.5$ | $76.4 \pm 5.6$ | $57.8 \pm 4.5$ |
| LSD | $99.8 \pm 0.4$ | $79.8 \pm 4.5$ | $70.0 \pm 0.6$ | $71.9 \pm 5.2$ | $69.8 \pm 8.8$ | $61.5 \pm 5.2$ |
| CSD | $97.4 \pm 3.4$ | $79.8 \pm 5.4$ | $64.0 \pm 6.2$ | $83.2 \pm 0.5$ | $96.2 \pm 9.1$ | $63.0 \pm 2.8$ |
| METRA | $92.8 \pm 4.2$ | $78.0 \pm 5.3$ | $54.8 \pm 9.5$ | $82.5 \pm 1.5$ | $83.4 \pm 7.5$ | $50.7 \pm 2.2$ |
| LEADS | $100.0 \pm 0.0$ | $\mathbf{96.0 \pm 4.3}$ | $\mathbf{90.8 \pm 5.3}$ | $89.7 \pm 8.8$ | $87.4 \pm 4.6$ | $\mathbf{85.4 \pm 7.2}$ |

Table 1: Final coverage percentages for each method on each environments. Bold indicates when a single method is statistically superior to all other methods ($p < 0.05$). Full T-test results are presented in Appendix B.

ones, which do not rely on skill diversity and rather rely on exploration bonuses: Random Network Distillation (RND) [7] and Never Give Up (NGU) [2].

A metric of state space coverage should characterize how widespread the state distribution is, in particular across behavior descriptors that are meaningful for the task at hand. Following the practice of previous work, for high-dimensional environments, we use a featurization of the state, retaining a low-dimensional representation of meaningful variables: the $(x, y, z)$ coordinates of the gripper in Fetch-Reach, the $(x, y)$ coordinates of the puck in Fetch-Slide, and the angles of the finger's two joints and the hinge's angle in Finger. This projection is then discretized and the coverage of a trajectory is defined as the number of visited cells. Figure 5 depicts the evolution of coverage, normalized by the maximum coverage achieved by any run in the current environment. This figure illustrates how coverage progresses relative to the number of samples taken in the environment since the beginning of the algorithm. Shaded areas represent the standard error across five seeds for each algorithm.

One can note that LEADS outperforms other methods across almost all tasks. This is especially notable in the Fetch-Slide environment, where LEADS exceeds the coverage of all other methods by over 20%. Furthermore, besides LEADS, no algorithm is consistently good across all tasks. In the Fetch-Reach task, DIAYN, METRA, and CSD each achieved a coverage between 80% and 85%. LEADS excelled with 90% coverage, significantly outperforming NGU at 50% and SMM and RND, both at 20%. CSD surpasses LEADS and all other methods on the Finger environment, achieving

a relative mean coverage of 96% where LEADS is still competitive with 87%. On this specific benchmark, NGU, METRA and LSD are competitive with LEADS, whereas DIAYN, SMM and RND exhibit low coverage.

## 5 Conclusion

In this work, we consider the problem of exploration in reinforcement learning, via the search for behavioral diversity in a finite set of skills. We take inspiration from the classical framework of maximization of the mutual information between visited states and skill descriptors, and illustrate why it might be insufficient to promote exploration. This motivates the introduction of a new objective function for skill diversity and exploration. This objective exploits an uncertainty measure to encourage extensive state space coverage. We leverage off-the-shelf estimators of the successor state representation to estimate this non-stationary uncertainty measure, and introduce the LEADS algorithm.

Specifically, LEADS estimates the successor state representation for each skill, then specializes each skill towards under-visited states while keeping skills in distinct state distributions. This results in an efficient method for state space coverage via skill diversity search. Our experiments highlight the efficacy of this approach, achieving superior coverage in nearly all tested environments compared to state-of-the-art baselines.

LEADS intrinsically relies on good SSR estimators. In this work, we used C-Learning [14] for all experiments, which proved efficient but might reach generalization limits in some environments. As a consequence, advances on the question of reliable SSR estimation will directly benefit LEADS, opening new perspectives for research.

## Acknowledgments

This work was performed using HPC resources from CALMIP (Grant 2016-[p21001]).

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

## A  A formal motivation for the uncertainty measure of Equation 7

In Section 3.2, we introduced a new optimization target, based on the uncertainty measure $u_t(s, z)$, used in LEADS to strategically distribute probability mass across states for each skill to foster effective exploration. $u_t(s, z)$ is introduced in Equation 7, which is reproduced here for clarity:

$$
u_t(s, z) = \underbrace{\log\left(\frac{m_t(s_0, s, z_i)}{\sum_{k=1}^{t-1}\sum_{z'} m_k(s_0, s, z')}\right)}_{\text{Explore under-visited areas}} + \underbrace{\sum_{z' \neq z} \log\left(\frac{m_t(s_{t-1}^z, s, z)}{m_t(s_{t-1}^{z'}, s, z')}\right) + \log\left(\frac{m_t(s_0, s, z)}{m_t(s_0, s, z')}\right)}_{\text{Repulsion between skills}}
$$

This section delves into the theoretical foundations of this target. We begin by discussing the left term of Equation 7, which corresponds to the exploration component of the measure. Subsequently, we detail the right term, which promotes greater repulsion between the skills.

### A.1  Exploration term (left side of Equation 7)

The Successor State Representation (SSR) extends the concept of the occupancy measure. Unlike the traditional occupancy measure, which begins with the initial state distribution, the SSR is an estimation of the occupancy measure starting from any state. In practice, we work with the expected sum of discounted presences of a state over time because we know this measure exists regardless of the ergodicity of the Markov process.

LEADS is an epoch-based algorithm with each epoch consisting of four phases, as presented in Section 3.2 and Algorithm 1:

1. Sample transitions from the environment for each skill,
2. Learn the SSR of each skill (for the given policy),
3. Define a state on which to put more probability mass for each skill,
4. Use the objective of Equation 6 to update the policy so that it implements this probability mass transport plan.

Since the policy changes in each epoch, the discounted occupancy measure of the policy also evolves over time. For a given skill and epoch $n$, it can be defined as:

$$
\rho_n^{\pi_z}(s) = \mathbb{E}_{\substack{s_0 \sim \delta_0 \\ a \sim \pi(\cdot|s_t, z) \\ s_{t+1} \sim P(\cdot|(s_t, a_t))}} \left[\sum_{t=0}^{\infty} \gamma^t \mathbb{1}(s_t = s)\right]
$$

This occupancy measure has a direct estimator, using the SSR in the initial state $s_0$:

$$
\rho_n^{\pi_z}(s) = m_n(s_0, s, z)
$$

At epoch $n$, the state occupancy measure of *all* skills is the mixture of all skills' state occupancy measures:

$$
\rho_n^{\pi}(s) = \sum_{\mathcal{Z}} p(z)\rho_n^{\pi_z}(s)
$$

Then, the left side term in the definition of the uncertainty measure $u_t(s, z)$ corresponds to the contribution of a state $s$ to the following Kullback-Leibler divergence:

$$
\text{KL}\left(\rho_n^{\pi_z} \parallel \sum_{k=1}^{n-1} \rho_k^{\pi}\right) = \mathbb{E}_{s \sim \rho_n^{\pi_z}}\left[\log\left(\frac{\rho_n^{\pi_z}(s)}{\sum_{k=1}^{n-1} \rho_k^{\pi}(s)}\right)\right] \tag{9}
$$

$$
= \int_{\mathcal{S}} \rho_n^{\pi_z}(s)\log\left(\frac{\rho_n^{\pi_z}(s)}{\sum_{k=1}^{n-1} \rho_k^{\pi}(s)}\right) \mathrm{d}s
$$

$$
= \int_{\mathcal{S}} \rho_n^{\pi_z}(s)\log\left(\frac{m_n(s_0, s, z)}{\sum_{k=1}^{n-1}\sum_{z'} m_k(s_0, s, z')}\right) \mathrm{d}s
$$

Note in particular that this divergence accounts for the discrepancy between $\rho_n^{\pi_z}$, ie. the states visited by skill $z$ at epoch $n$, and $\sum_{k=1}^{n-1} \rho_k^{\pi}$, ie. the states previously explored by *all* skills in *all* past epochs. Thus, the KL divergence of Equation 9 serves as a measure of novelty for the current distribution. Intuitively, the newer a state visited by the current distribution $\rho_n^{\pi_z}(s)$ is, the larger its contribution to this KL divergence will be. Recall that at this stage, we do not modify the policy (it is the role of the objective function of Equation 6), but rather aim at identifying which states within the support of $\rho_n^{\pi_z}$ should be those skill $z$ is encouraged to explore next. By defining the state-wise novelty score $\log\left(\frac{m_n(s_0,s,z)}{\sum_{k=1}^{n-1}\sum_{z'} m_k(s_0,s,z')}\right)$, the distribution $\delta(s|z)$ unfolds as the Dirac distribution which maximizes the expected value of this score. Overall, LEADS aims to identify and prioritize states with the highest novelty score for each skill, thereby enhancing exploration by shifting each skill's induced distribution towards these novel states.

## A.2 Repulsion term (right side of Equation 7)

Building on the previous discussion about the formal motivation of the exploration term on the left side in 7, we now explore the right side term, which is intended to enhance repulsion between skills.

The right side term of Equation 7 can be interpreted as the contribution of a state $s$ within a sum of multiple Kullback-Leibler divergences:

$$\sum_{z'\neq z} \underbrace{\mathrm{KL}(m_n(s_{n-1}^z,\cdot,z) \| m_n(s_{n-1}^{z'},\cdot,z'))}_{(a)} + \underbrace{\mathrm{KL}(\rho_n^{\pi_z}(s) \| \rho_n^{\pi_{z'}}(s))}_{(b)} \tag{10}$$

In this equation, term (a) represents the KL divergence of the occupancy measure for the current skill $z$, starting from its last target state, compared to the occupancy measure of each other skill starting from their respective last target states. Maximizing this KL divergence aims to make the distribution of each skill (starting from their last target state) as distinct as possible. Similarly, term (b) is the KL divergence between the occupancy measures of a skill $z$ and that of any other skill $z'$. Maximizing these two KL divergences leads to disjoint distributions for each skill. As in the previous subsection, we do not change the policy at this stage (this would lead to an ill-defined optimization problem as the KL can become unbounded), but rather aim at identifying which are the states that contribute most to these KL divergences, so that we can promote exploration towards them. Overall, LEADS combines all these terms into the uncertainty measure of Equation 7 to determine which state contributes the most to these KL divergences for each skill. In turn, these states define the distribution $\delta$ designed to promote exploration, by increasing the weight of the most likely states of $\delta$ in Equation 6

## A.3 Numerical advantage of Lower Bound 4

This section offers a more detailed explanation of the passage from the definition of Mutual Information 2 to the Lower Bound 4. The mutual information can be defined as: $\mathbb{E}_{(s_2,z)\sim p(s,z)}\left[\log\left(\mathbb{E}_{s_1}\left[m(s_1,s_2,z)\right]\right)\right]$, with $s_1 \sim p(s_1|z)$.

Suppose we wish to maximize this quantity using SGD. For each sample $(s_2,z)$, we need to compute the expectation $\mathbb{E}_{s_1\sim p(s|z)}\left(m(s_1,s_2,z)\right)$ (because the log is not linear). In practice, this is computationally expensive because it requires sampling a mini-batch of $s_1$ states for every single $s_2$ state. Instead, we use Jensen's inequality, leveraging the concavity of the log function to derive the lower bound:

$$\mathbb{E}_{z\sim p(z)\ s_2\sim p(s|z)\ s_1\sim p(s|z)}\left[\log(m(s_1,s_2,z))\right]$$

Maximizing this new quantity can be done with SGD by sampling three independent mini-batches at each iteration.

# B Supplementary Analyses and Visualizations

LEADS is based on the estimation of the successor state representation for each skill. Figure 6 and Figure 7 show the final SSR and uncertainty measure, respectively, for the maze and Fetch tasks. These correspond to the same runs shown in Figure 2 for LEADS.

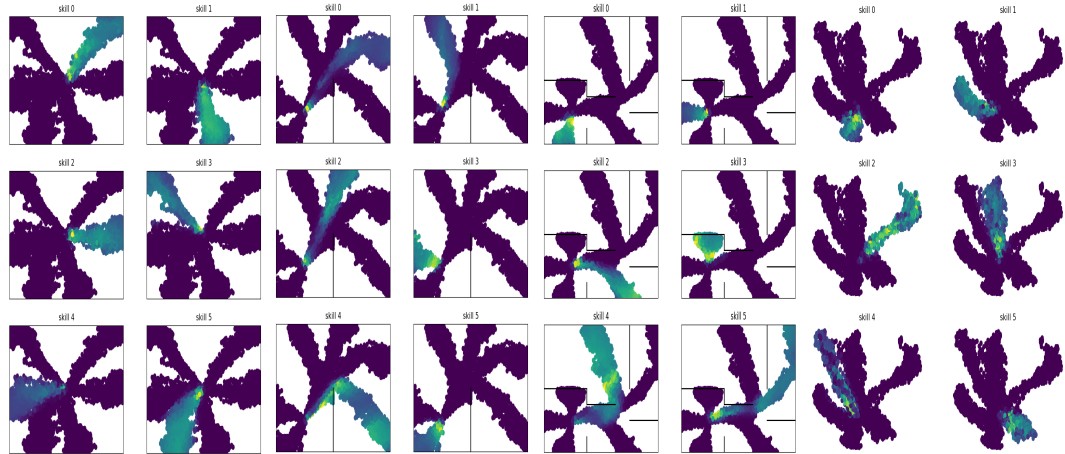

Figure 6: The SSR $m(s_0, s, z)$ at a given epoch during training on each task, per skill, normalized in $[0, 1]$.

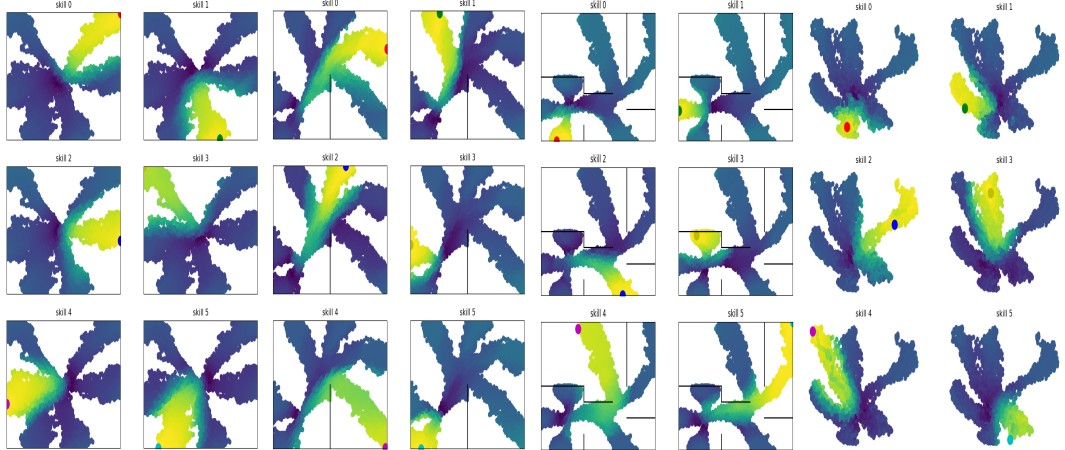

Figure 7: The uncertainty measure $u(s, z)$ at a given epoch during training on each task, per skill, with the maximum state highlighted.

Figure 5 illustrates the temporal evolution of coverage for all methods across various environments, evaluated over five different seeds. We extend our analysis by presenting the p-values from the t-tests conducted on the distributions of the final (highest) coverage values for these five seeds, comparing the first method to both the second and third methods. We conducted a t-test for each environment in which no method outperformed the second method by a margin greater than 10%. The results are visible in the table 2.

| Environment: Method 1/ Method 2 | p-value |
|---|---|
| Umaze: LEADS/CSD | 0.01522 |
| Umaze: LEADS/LSD | 0.01152 |
| Hard: LEADS/CSD | 0.00246 |
| Hard: LEADS/LSD | 0.00977 |
| Fetch Reach: LEADS/CSD | 0.06354 |
| Fetch Reach: LEADS/DIAYN | 0.08307 |
| Finger: CSD/LEADS | 0.07072 |
| Finger: CSD/LSD | 0.00753 |
| Fetch Slide: LEADS/CSD | 0.00070 |
| Fetch Slide: LEADS/LSD | 0.00055 |

Table 2: Comparison of p-values for paired t-tests across methods

## B.1 Key Differences with Baseline Methods

The LEADS algorithm learns diverse skills that evolve over time to maintain exploration of the state space. Unlike the baselines, LEADS does not rely on any reward function to be maximized. Instead, it uses the same tool, the SSR, to plan exploration and adjust the set of skills to ensure effective exploration. Unlike static skills, the skills learned by LEADS are dynamic, continuously adapting throughout the algorithm's execution to move away from previously explored areas, as explained in A. LEADS achieves this by maximizing an uncertainty measure, which includes elements from the Kullback-Leibler divergence between the current occupancy measure of each skill and the history of occupancy measures from the set of skills.

This mechanism differs from algorithms such as DIAYN [13], LSD [33], CSD [34], and METRA [35], which aim to learn a static set of skills. These methods evolve the skills until reaching an optimum or sub-optimum. DIAYN maximizes mutual information, represented as the Kullback-Leibler divergence between $\mathbb{P}(S, Z)$ and $\mathbb{P}(S)\mathbb{P}(Z)$, while LSD, CSD, and METRA maximize the Wasserstein distance between these distributions using different metrics.

In contrast, NGU [2] and RND [7] use uncertainty methods that evolve only a single policy over time and do not seek to learn a set of diverse skills. The method closest to LEADS is SMM [26], which also learns a set of diverse skills that explore using a specific uncertainty measure. SMM uses the loss from a VAE [17] as the uncertainty measure, framing the optimization as a two-player adversarial game.

LEADS uses a novelty-based uncertainty measure, which appears to lead to a more stable optimization process leading to a more continous and efficient exploration.

## C  Hyperparameters

The following table (Table 3) summarizes the hyperparameters used in our experimental setup.

| Hyperparameter | Value |
|---|---|
| $n_{\text{skill}}$ | 6 |
| $z_{\text{dim}}$ | 20 |
| $\lambda_h$ | 0.05 |
| $\gamma$ | 0.95 |
| $\lambda_{\text{c-learning}}$ | 0.5 |
| $\alpha_\theta$ | $5 \times 10^{-4}$ |
| $\alpha_{\text{c-learning}}$ | $5 \times 10^{-4}$ |
| $n_{\text{episode}}$ | 16 |
| $n_{\text{SGD, c-learning}}$ | 256 |
| $n_{\text{SGD, actor}}$ | 16 |
| $n_{\text{archive}}$ | 1 |
| batch size$_{\text{c-learning}}$ | 1024 |
| batch size$_{\text{loss}}$ | 1024 |

Table 3: Hyperparameters used for LEADS

| Layer | Type | Input Dimensions | Output Dimensions | Activation |
|---|---|---|---|---|
| | | **Classifier Network** | | |
| 1 | Dense | (State Dim)$\times$2 + Action dim + Z Dim | 256 | ReLU |
| 2 | Dense | 256 | 128 | ReLU |
| 3 | Dense | 128 | 1 | Sigmoid |
| | | **Actor Network** | | |
| 1 | Dense | State Dim + Action Dim | 256 | ReLU |
| 2 | Dense | 256 | 256 | ReLU |
| 3 | Dense | 256 | Action Dim | Linear |
| 4 | Dense | 256 | Action Dim | Tanh |

Table 4: Structure of the Classifier and Actor Networks

## C.1 The number of skills

As with other skill-based algorithms, LEADS's coverage changes with the number of skills. However, the skills learned by LEADS evolve throughout the entire training procedure to visit unexplored areas. Hence, one can expect LEADS to require fewer skills to achieve the same final coverage performance than algorithms that learn static skills, like DIAYN [13]. Choosing the number of skills involves a tradeoff between the acceptable computational runtime of the algorithm (which increases with the number of skills) and the desired efficiency in state space coverage. This choice is also highly problem-dependent. Dynamic adaptation of the number of skills is a challenging and open topic, covered by works such as the one proposed by Kamienny et al. [24]. Although this remains an open question, for LEADS we worked with a hand-chosen, predetermined number of skills and did not focus on this aspect.

## D Failure cases of the successor state representation estimation

LEADS relies intrinsically on the quality of the SSR estimator. When this estimator is poor, LEADS might not yield diverse and exploratory skills. For instance, in a number of standard MuJoCo environments from the Gymnasium suite [43], LEADS performs poorly as a consequence of C-Learning's inability to obtain a reliable estimation of the successor state representation. Figures 8 and 9 illustrate this phenomenon on the HalfCheetah benchmark after 25 epochs of LEADS. We evaluate the discrepancy between the estimated state occupancy measure and the actual distribution of states for a set of 6 skills. To enable visualization of these quantities, we project these densities on the $(x, y)$ plane where $x$ and $y$ are the coordinates of the center of mass of the cheetah's torso. Figures 8 reports the estimated state occupancy measure from the initial state $m(s_0, \Pi(s), z)$, where $\Pi(s) = (x, y)$, using the SSR. In contrast, Figure 9 displays the scatter plot of visited states for each skill in the $(x, y)$ plane. Despite experimenting with an extensive range of hyperparameters for C-Learning, a satisfactory estimation of the measure could not be obtained and the probability density of Figure 8 could not match the $(x, y)$ coordinates of the visited states of Figure 11.

As a comparison, Figures 10 and 11 illustrate the identical experiment conducted in the Fetch-Reach environment reported in the main body of the paper. The state occupancy estimated with the SSR of Figure 10 matches closely the states encountered by each skill in Figure 11. This corroborates the quantitative results reported in Figure 5: when a reliable estimation of the SSR is available, LEADS performs effectively.

Investigating why C-Learning performs poorly in such MuJoCo environments is beyond the scope of this paper. One could conjecture, following the conclusions of Blier et al. [4], that learning a successor state representation that generalizes efficiently through high-dimensional state spaces is a challenging task. Although we could not obtain conclusive results with C-Learning in these specific cases, recent methodologies [15, 45] demonstrate improved generalization capabilities. These new methods appear to be more successful in MuJoCo environments. We reserve the extension of LEADS to such SSR estimators for future work.

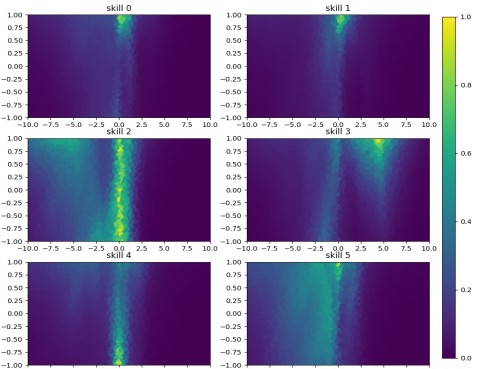 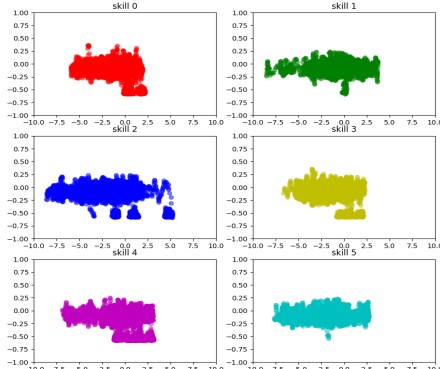

Figure 8: The SSR $m(s_0, \Pi(s), z)$ at epoch 25 for each skill in HalfCheetah environment

Figure 9: (x,y) coordinates of the states sampled by each skill for 16 rollouts each in HalfCheetah environment

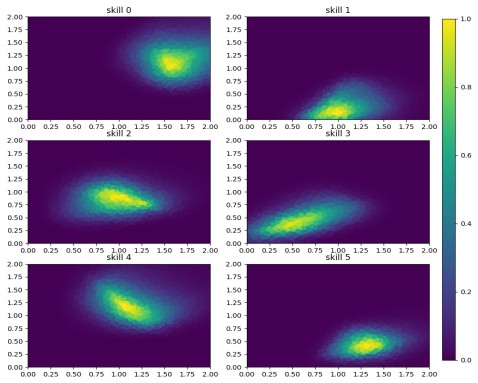 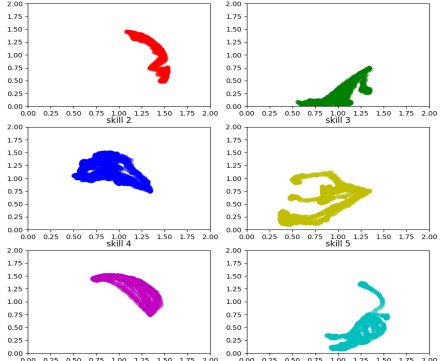

Figure 10: The SSR $m(s_0, \Pi(s), z)$ at epoch 25 for each skill in FetchReach environment

Figure 11: (x,y) coordinates of the states sampled by each skill for 16 rollouts each in FetchReach environment

