# OpenReview forum: "Exploration by Learning Diverse Skills through Successor State Representations"
_NeurIPS.cc/2024/Conference — NeurIPS 2024 poster_

### Official Review · Reviewer_8kZQ · 2024-07-09

**Soundness:** 3
**Presentation:** 4
**Contribution:** 3
**Rating:** 7
**Confidence:** 4

**Summary:**

This paper proposes a new skill-based exploration method that leverages successor states. The authors start by arguing the inadequacy of maximizing mutual information as objective for learning diverse skills that also encourage exploration. Then the authors define the mutual information in terms of successor state measures to derive a novel uncertainty measure that ranks states highly that encourage visitation within the current skill but that other skills haven’t visited.  The authors provide comprehensive experiments showing that they outperform common intrinsic motivation methods and other well known skill diversity exploration methods.

**Strengths:**

- The presentation of the paper is excellent. It is easy to read and follow.
- The authors motivate the paper very well and start from existing work to derive a novel method.
- The experiments seem fitting although somewhat simple.
- The authors perform thorough ablations and also analyse failure cases.

**Weaknesses:**

- The first 2-D environment seems a bit simple. Although I think it is adequate to demonstrate exploration behaviour, it would be interesting to see how LEADS performs with more complex observations or tasks.

**Questions:**

- have you tested your methods on any rgb observations? It would be interesting to see if C-Learning is still robust in this case.
- Could you think of any way to perform C-Learning in high dimensional environments? Is the bottleneck related more to the task complexity or observational complexity?

**Limitations:**

The authors have addressed the limiting components of their method.

---

> ### Author Rebuttal · Authors · 2024-08-07
>
> # Answer: Reviewer 8kZQ
> We thank the reviewer 8kZQ for their very positive comments on the paper presentation and for their insightful comments on C-learning. We hope to address their questions in our response.
>
> We did find that the application of C-learning is limited based on the dimensionality of the problem. The application of C-learning to the high-dimensional state spaces of the MuJoCo environments was a challenging aspect of this work, as detailed in Appendix D. We therefore did not try RGB environments, as this would require further fine-tuning of C-learning on a convolutional architecture. However, recent contributions [1][2] building upon C-Learning illustrate that learning this measure in such environments is feasible, making this application an interesting direction for future work. We will include these references in the discussion on the limitations of C-learning in Appendix D.
>
> [1] Eysenbach, Benjamin, et al. "Contrastive learning as goal-conditioned reinforcement learning." Advances in Neural Information Processing Systems 35 (2022): 35603-35620.
> [2] Zheng, Chongyi, Ruslan Salakhutdinov, and Benjamin Eysenbach. "Contrastive Difference Predictive Coding." The Twelfth International Conference on Learning Representations.

---

> > ### Comment · Reviewer_8kZQ · 2024-08-11
> >
> > Thank you for the answer!
> >
> > I think it would be interesting to try these paradigms on problems that do not require immediate skill learning such as in MuJoCo, but where skill also goes through many hierarchies. For example, the agent needs to learn how to pick up an object on a more detailed level, but also needs to understand how to use that object in a broader context. I do realise this is out of the scope for the paper.
> >
> > Overall, I think this is a nice paper, with a simple and clear idea.

---

> > > ### Author Response · Authors · 2024-08-12
> > > **Official Comment to Reviewer 8kZQ**
> > >
> > > Thank you for your positive feedback and insightful suggestion. We will certainly take this direction into account for future research. We appreciate your valuable input and thank you for taking the time to review our paper.

---

### Official Review · Reviewer_axfE · 2024-07-11

**Soundness:** 3
**Presentation:** 2
**Contribution:** 2
**Rating:** 6
**Confidence:** 2

**Summary:**

The paper introduces LEADS, an algorithm to learn diverse skills that additionally encourages exploration. LEADS is motivated by the observation that common mutual information-based diversity-seeking algorithms cannot effectively encourage exploration. LEADS instead proposes a new objective based on successor state measures that explicitly encourage the coverage of state spaces. Experiments conducted on state-based control problems validate the effectiveness of LEADS in learning diverse skills that better cover the whole state space than baselines.

**Strengths:**

1. The work is well motivated with both illustrative examples and theoretical analysis.
2. This work provides extensive comparisons with baselines. The qualitative visualization is particularly helpful for readers to understand the learned exploratory behavior of the proposed algorithm.

**Weaknesses:**

1. In the experiment section, it would be better to briefly introduce the compared baseline methods and point out the main difference between LEADS and these methods before delving into the detailed discussion of results.
2. Besides better coverage of states, are there other advantages of learning more exploratory behaviors? For example, can LEADS achieve better success rates on hard-exploration problems?
3. How does LEADS perform with a different number of skills? Is there a best choice of the number of skills or is LEADS insensitive to this hyper-parameter?

**Questions:**

See the weaknesses part.

**Limitations:**

LEADS relies on the state occupancy measure estimator to learn diverse skills and encourage exploration. For behaviors that cannot be easily distinguished from the state occupancy measure, the algorithm might not be able to handle them properly.

---

> ### Author Rebuttal · Authors · 2024-08-07
>
> # Answer: Reviewer axfE
>
> We thank Reviewer axfE for their comments. In our general response, we address their specific concern regarding a more comprehensive comparison of LEADS with other baselines. Below, we will address the remaining questions and concerns about our study.
>
> ### Beyond the maximum coverage objective:
> We appreciate the reviewer raising this insightful question, which opens up important avenues for discussion.
> As supported by multiple studies [1][2][3][4], exploration represents a critical step in deriving effective policies, especially in hard exploration problems. We conjecture that initializing, for example, a goal-reaching algorithm using the replay buffer obtained via LEADS would yield better success rates, particularly in hard exploration problems. For instance, in Figure 2, in the Hard-maze environment, the transitions sampled by skill 4 are essential for training any goal-reaching policy over goals in that part of the maze. We also conjecture that another expected advantage could be the design of reusable/composable skills in a hierarchical setting, but we consider it out of the scope of this study and reserve it for future work. We propose mentioning this direction in the conclusion section.
>
> ### Different number of skills:
> As with other skill-based algorithms, LEADS's coverage changes with the number of skills. However, the skills learned by LEADS evolve throughout the entire training procedure to visit unexplored areas. Hence, one can expect LEADS to require fewer skills to achieve the same final coverage performance than algorithms that learn static skills, like DIAYN. Choosing the number of skills involves a tradeoff between the acceptable computational runtime of the algorithm (which increases with the number of skills) and the desired efficiency in state space coverage. This choice is also highly problem-dependent. Dynamic adaptation of the number of skills is a challenging and open topic, covered by works such as [5]. Although this remains an open question, for LEADS we worked with a hand-chosen, predetermined number of skills and did not focus on this aspect, as is common in most works in the literature. We propose including this discussion in Appendix C on Hyperparameters.
>
>
> We hope these answers address the concerns of the reviewer and lead to further exchanges.
>
> [1] Deepak, Pulkit et al. Curiosity-driven exploration by self-supervised prediction. In International conference on machine learning, pages 2778–2787. PMLR, 2017.
>
> [2] Yuri, Harrison et al. Exploration by random network distillation. arXiv preprint arXiv:1810.12894, 2018.
>
> [3] Adrià, Pablo et al. Never give up: Learning directed exploration strategies. arXiv preprint arXiv:2002.06038, 2020.
>
> [4] Zhaohan, Shantanu, Miruna et al. BYOL-explore: Exploration by bootstrapped prediction. Advances in neural information processing systems, 35:31855–31870, 2022.
>
> [5] Kamienny, Tarbouriech et al.(2022, April). Direct then Diffuse: Incremental Unsupervised Skill Discovery for State Covering and Goal Reaching. In ICLR 2022.

---

> > ### Comment · Reviewer_axfE · 2024-08-11
> >
> > Thank you for your answer. My questions about further advantages beyond maximum coverage and the number of skills are mostly addressed. I would like to raise the score to 6.

---

> > > ### Author Response · Authors · 2024-08-12
> > > **Official Comment to Reviewer axfE**
> > >
> > > Thank you for your response and for raising the score. We appreciate your feedback, which has been instrumental in improving the readability of our paper. Thank you for your time and consideration.

---

### Official Review · Reviewer_98FT · 2024-07-12

**Soundness:** 3
**Presentation:** 3
**Contribution:** 3
**Rating:** 7
**Confidence:** 2

**Summary:**

The authors aim to solve the problem of exploring to learn a diverse set of skills. To do this, they modify a commonly used mutual information objective in two ways: first, they apply it to the successor measure, and second, they change the sampling distribution to focus on states with high uncertainty. They test this method on a number of challenging goal-conditioned environments, and show that it performs better than existing methonds on several environments, as well as being the only method which achieves good state coverage on all environments.

**Strengths:**

Originality:
		There are two main original contributions to this method: the application of mutual information to state successor measures, and the substitution of the uncertain measure as the sampling distribution. In my opinion, this is sufficient originality.

	Quality:
		The derivation of the objective is correct. The authors test on a varied set of environments in challenging high dimensional spaces.

	Clarity:
		The work is presented well and is easy to follow.

	Significance:
		The work is a decent improvement over SOTA the authors compare to.

**Weaknesses:**

Using the uncertainty measure instead of the actual distribution breaks most of the theoretical properties this algorithm may have had, but I can see why it was done. It would be nice if there were a more solid mathematical interpretation of this objective, such as a tradeoff between an explicit exploration objective and the mutual information objective, but I understand this is not always possible.

**Questions:**

What hand environments are being used? I assume this is HandReach?

**Limitations:**

The authors adequately address the limitations of their work

---

> ### Author Rebuttal · Authors · 2024-08-07
>
> # Answer: Reviewer 98FT
> We thank Reviewer 98FT for their comments. In the following, we address their specific questions and concerns about our study.
>
> ### Hand environment:
> Indeed, the environment used in our tests is HandReach. We will make this clear in the experimental section.
>
> ### Theoretical study of LEADS
>
> We agree with the reviewer's comment about the benefits of a deeper theoretical understanding of the LEADS algorithm and specifically the interplay between the mutual information objective and an explicit exploration objective. Through this study, we aim to demonstrate that mutual information maximization can lead to exploration when coupled with an appropriate objective, such as the maximization of the uncertainty measure we propose, along with a sufficient successor state measure. A theoretical understanding of this interplay motivated the writing of Appendix A, which we hope serves as a starting point for this discussion. More specifically, the analysis of the exploration term in Equation 7, detailed in Appendix A.1, draws a link between the uncertainty measure we use and a specific Kullback-Leibler divergence, providing a practical understanding of its maximization.
>
> In future work, we hope to build upon this theoretical insight to derive more formally motivated objectives based on this study.

---

> ### Comment · Reviewer_98FT · 2024-08-12
>
> Thank you for your response.
>
> The connection to the KL divergence does help motivate the method a bit better. I have increased my score to a 7

---

> > ### Author Response · Authors · 2024-08-13
> > **Official Comment to Reviewer 98FT**
> >
> > We thank you for your feedback and for the time you invested in reviewing our paper.

---

### Official Review · Reviewer_QVxs · 2024-07-14

**Soundness:** 2
**Presentation:** 2
**Contribution:** 3
**Rating:** 7
**Confidence:** 4

**Summary:**

Having an agent learn a set of diverse skills potentially affords the agent better environment exploration. A lack of diversity among the learned skills may reduce the agent's ability to discover newer informative states in the environment. This paper formalizes a method for an agent to learn a set skills by enhancing previous defined mutual information between the states and skills so that it explicitly encourages diversity in exploration. They then evaluate their approach (LEADS) against other algorithms in a number of domains to demonstrate its ability to more effectively explore the state space.

**Strengths:**

**Originality**
While the paper does not introduce any new concept or problem, I consider that the paper does propose an interesting variation to the use of mutual information between states and skills to learn exploratory options that maximize coverage of the state space at the same time that the options are diverse. Both successor features and using mutual information for exploration existed in the literature before this work. The combination done in this way seems interesting.

**Significance**
The algorithm represents progress towards the solution of a very relevant problem in deep reinforcement learning: continual exploration based on online observations of the state space.

**Clarity**
The explanation of the problem and the general idea of the solution and the paper are easy to follow. However there are some issues explained later.

**Quality**
The intuitive concept behind combining successor state measures and mutual information is sound. While the resultant algorithm makes sense, I do not think it works necessarily how the authors explain. The experiments performed are reasonable for empirically backing their claims but the results were not analyzed properly (more later)

**Weaknesses:**

* While the explanation of the problem and the general idea of the solution are easy to follow. There is some mathematical imprecision in Section 3:
   1. A measure is a function from a $\sigma$-algebra to non-negative reals. Here the measure is defined for states rather than sets of states. This is more accurately the successor representation.
   2. The letter $p$ is used to denote multiple different probability density functions. This is confusing since, $s_1$ and $s_2$ correspond to different random variables: $p(s_1|z)$, the effective state distribution observed in the replay buffer is different from $p(s_2|z)$, which is the state distribution resulting from the discounted visitation.

* Besides the lack of mathematical precision, there is a statement that is not backed up by any argument: line 163 states that the lower bound results in eliminating the natural interplay between the skills that is embedded in the mutual information. There is no explanation for this assertion.

* The related works does not reference work on options or Eigenoptions which are very closely related do successor representations.

* The t-test seems an inappropriate statistical test --- Paired t-tests are better than t-test for comparing algorithms. (See https://arxiv.org/pdf/2304.01315#page=29)

* Sample standard deviation (as used in Fig. 5) is a measure of the variability of a distribution NOT a confidence interval over a mean. Statements such as "One can note that LEADS outperforms other methods across almost all tasks"  are unsubstantiated by the provided evidence. One should instead use standard error (as is commonly done in the literature) to obtain a confidence interval. However, a more appropriate choice would either by student-t confidence interval or a bootstrap confidence interval. Please see: https://arxiv.org/pdf/2304.01315#page=13

**Questions:**

* Notation is imprecise. E.g. what is the domain of policy? $\pi_\theta: \mathcal{S} \times \mathbb{R}^d \rightarrow \mathcal{A}$ or  $\pi_\theta: \mathcal{S}  \rightarrow \mathcal{A}$​​?

* The problem setting of Fig 1 is confusing. Is each state reachable? If so the mutual information calculation is wrong. If we assume only the states denoted by the skills are reachable then the calculation is correct but this assumption should be explicitly stated.
Treatment of how hyperparameters were handled was not specified. Were baselines tuned? How was tuning done? (E.g. grid search? random search? etc.)

* The first bound was introduced to remove the need to sample a minibatch $s_1$ from $z$, but this is still the case after the introduction of the bound. What was gained out of the replacement?

**Limitations:**

N/A.

---

> ### Author Rebuttal · Authors · 2024-08-07
>
> # Answer: Reviewer QVxs
>
> We thank Reviewer QVxs for their comments. In our general response, we address several of the concerns raised by the reviewer. Below, we specifically address their additional questions and concerns about our study.
>
> ### Statement in line 164:
>
> The sentence "natural interplays between skills" in the text refers to the interaction or coordination required among the different skills during the maximization of Mutual Information. Specifically, the reversed form of Mutual Information (MI) is:
>
> $$
> \mathcal{I}(S,Z) = \mathbb{E}_{\substack{z \sim p(z) \\ s \sim p(s|z)}} \left[\log \left(\frac{p(z|s)}{p(z)}\right)\right].
> $$
>
> To maximize this quantity, the distribution $p(z|s)$ in the log term must be as concentrated as possible on the skill $z$ that was used to sample this state. Therefore, the probability of any other skill on that state must be minimal. The term "natural interplays" thus informally describes the necessary coordination between skills to ensure that each skill is uniquely associated with specific parts of the environment, leading to maximal MI.
>
> In contrast, maximizing the lower bound given in Equation 4:
>
> $$
> \mathbb{E}_{\substack{z \sim p(z) \\ s_2 \sim p(s|z) \\ s_1 \sim p(s|z)}} \left[\log(m(s_1, s_2, z))\right]
> $$
>
> corresponds to increasing the probability mass on state $s_2$ (sampled in $p(s|z)$) starting from $s_1$ (also sampled in $p(s|z)$) for the skill $z$, independently of the probability mass of other skills on these two states. The maximization of this quantity does not account for the interactions between skills.
>
> In Equation 5, we derive a new lower bound that reintroduces these interplays:
>
> $$
> \underset{\substack{z \sim p(z) \\ s_2 \sim p(s|z) \\ s_1 \sim p(s|z)}}{\mathbb{E}} \left[\log \left(\frac{m(s_1, s_2, z)}{1 + \sum_{z' \in \mathcal{Z}} m(s_1, s_2, z')}\right)\right]
> $$
>
> In this formulation, the interplays between skills are captured in the denominator of the log term. To maximize this log expression, the denominator must be minimal. This occurs when the probability of transitioning to state $s_2$ from state $s_1$ (which were sampled using skill $z$) is minimal for all other skills $z'$.
>
> To conclude, the term "natural interplays" refers to whether the relationships and interactions between skills are preserved in the maximization process of these expectations. We propose to include the above reasoning as an explanation of the phrase in a new short appendix section.
>
> ### Eigenoptions in related work:
>
> In the study of Eigenoptions, the authors derive a way to decompose a given task into different options for which they propose automatic learning. We agree that the study could be relevant in the related work subsection "Successor features for MI maximization". We will include the following reference [1] in the final version of the paper.
>
> ### Statistical Analysis of the Results:
>
> We thank the reviewer for their suggestions and will modify the results in the final section. Specifically, we will use Standard Error and a Paired t-test in our final results. Below, we include p values using the paired t-test, which we will add to Appendix B. We note that the significance indication in Table 1 does not change when using the paired t-test; LEADS is significantly superior to all other methods on the Umaze, Hard, and Fetch Slide environments, and no other method significantly outperforms all other methods.
>
> | Method                     |   pvalue       |
> |----------------------------|----------------|
> | Umaze :  LEADS/CSD         |     0.01522    |
> | Umaze :  LEADS/LSD         |     0.01152    |
> | Hard :  LEADS/CSD          |     0.00246    |
> | Hard :  LEADS/LSD          |     0.00977    |
> | Fetch Reach :  LEADS/CSD   |     0.06354    |
> | Fetch Reach :  LEADS/DIAYN |     0.08307    |
> | Finger :  CSD/LEADS        |     0.07072    |
> | Finger :  CSD/LSD          |     0.00753    |
> | Fetch Slide :  LEADS/CSD   |     0.00070    |
> | Fetch Slide :  LEADS/LSD   |     0.00055    |
>
> ### Hyperparameter tuning:
>
> We propose the inclusion of the following statement in the article: "Hyperparameters were determined through manual testing on the Hard-Maze environment. Resulting hyperparameters are displayed in Table 3. Default values from the literature were used as hyperparameters for other methods." We include hyperparameters of all methods in the included code (https://anonymous.4open.science/r/LDS-BE6B).
>
> ### Figure 1 clarification:
>
> In the Figure 1, only the states denoted by the skills are reachable.
> We propose making the unreachable states gray and indicating this in the figure explanation.
>
> ### Clarification on the Introduction of the First Bound and its Benefits:
>
> We believe the reviewer refers to the use of Jensen's inequality to derive the first lower bound in Equation 4. The mutual information can be defined as: $\mathbb{E}_{(s_2, z) \sim p(s, z)} \left[ \log \left( \mathbb{E}s_1 \left[ m(s_1, s_2, z) \right] \right) \right]$, with $s_1 \sim p(s_1|z)$.
>
> Suppose we wish to maximize this quantity using SGD. For each sample $(s_2,z)$, we need to compute the expectation $\mathbb{E}_{s_1 \sim p(s|z)} \left( m(s_1,s_2,z) \right)$ (because the log is not linear). In practice, this is computationally expensive because it requires sampling a mini-batch of $s_1$ states for every single $s_2$ state. Instead, we use Jensen's inequality, leveraging the concavity of the log function to derive the lower bound:
>
> $$
> \mathbb{E}_{\substack{z \sim p(z) \\ s_2 \sim p(s|z) \\ s_1 \sim p(s|z)}}\left[\log(m(s_1,s_2,z))\right]
> $$
>
> Maximizing this new quantity can be done with SGD by sampling three independent mini-batches at each iteration. We propose including the above explanation in a new appendix section that expands on section 3.1
>
> [1] Machado, M. C., Rosenbaum, C., Guo, X., Liu, M., Tesauro, G., & Campbell, M. (2018, February). Eigenoption Discovery through the Deep Successor Representation. In International Conference on Learning Representations.

---

> > ### Comment · Reviewer_QVxs · 2024-08-12
> > **Response to rebuttal**
> >
> > I would like to thank the authors for their response. The changes made to paper will definitely improve it and they address my major concerns. I will be increasing my score.

---

> > > ### Author Response · Authors · 2024-08-13
> > > **Official Comment to Reviewer QVxs**
> > >
> > > Thank you for your response. We appreciate your acknowledgment of the changes made and are glad to hear that they address your major concerns. Your feedback has been instrumental in improving the paper. Thank you for your thorough review and valuable suggestions.

---

### Author Rebuttal · Authors · 2024-08-07

We thank all the reviewers for their insightful feedback, which has been crucial in refining our paper and pinpointing areas that require further clarification. In this general response, we address the concerns raised by reviewers regarding the paper's clarity. We believe these concerns will lead to the main changes in the text, and we include them in the general response to encourage further discussion if needed. Additionally, we address all remaining concerns raised by each reviewer in the respective rebuttal sections.


* **The use of the word measure in "Successor State Measure":**


(reviewer QVxs)
We agree that the term "Successor State Measure" can be misleading and that the Successor State (Representation in the more generic case) of a policy does not adhere to the strict definition of a measure. Considering a measurable space $(\mathcal{S}, \Sigma)$, where $\mathcal{S}$ is a set and $\Sigma$ is a $\sigma$-algebra on $\mathcal{S}$, our definition of Successor State Measure indeed refers to the density of a measure defined on that measurable space. Depending on the reviewer’s viewpoint, we propose to either clarify this in the definition of the Successor State Measure or to replace "Successor State Measure" with "Successor Representation" throughout the text.

* **The use of $p$ to denote distributions:**


(reviewer QVxs)
The reviewer’s remark is correct. In Equation 3, we use the law of total probability to obtain: $p(s_2|z) = \mathbb{E}_{s_1 \sim p(s|z)}\left[ p(s_2|s_1,z) \right]$. We then use our estimation of the Successor State as an approximation of $p(s_2|s_1,z)$. More specifically, just like the density of the random variable $s_1$ (the replay buffer) is an empirical approximation of the density $p(s|z)$ obtained via a Monte Carlo process, the density of the random variable $s_2$ is an approximation of $p(s|z)$ obtained by approximating our Successor State as $p(s_2|s_1, z)$ in the expectation. We could indeed propose new notations in the text, but we believe it would be more appropriate to explicitly clarify these approximations in the text to avoid overly complex notations.

* **Policy mapping:**


(reviewer QVxs)
The policy's domain is $\mathcal{S} \times \mathbb{R}^d$ and its image by the policy $\pi$ is the set of distributions over the action space $\Delta(\mathcal{A})$. We will make this more explicit in the text.

* **Providing a better distinction with other baselines:**


(reviewer axfE)
Due to lack of space, including such details and comparison was not possible in the main text of the paper. But we agree it would make the paper easier to read. We have added an Appendix to better cover the baseline methods and their differences with LEADS, and referred to it at the beginning of the experimental section in the main text.

---

### Decision · Program_Chairs · 2024-09-25

**Decision:**

Accept (poster)

**Comment:**

This paper received four positive reviews with minor points that were all clarified in the author's rebuttal.

All reviewers agreed the idea proposed in this work (combining successor representations in the mutual information objective typically used in skill learning) is original and significant. There is also consensus that the proposed algorithm is correct. The experimental results are convincing and compare with seven related methods in many scenarios. The paper is also clearly presented.

While intuitively well motivated, the algorithm lacks a detailed theoretical analysis, and it is only motivated formally in Appendix A. However, I believe that the empirical part is strong enough to recommend acceptance.

The authors are encouraged to incorporate the following suggestions in the revised version.

- Incorporate all the points addressed in the rebuttal, including Figure 1 clarification, and the sentences mentioned in the response to reviewer QVxs.

- The authors should fix the mathematical and notational imprecisions mentioned in the rebuttal phase.

- It is recommended to replace the term "measure" with "representation".

- Incorporate relevant missing related work, such as [1].

- Include the discussion regarding the choice of the number of skills and the existing trade-offs.


[1] Machado, M. C., Rosenbaum, C., Guo, X., Liu, M., Tesauro, G., & Campbell, M. Eigenoption Discovery through the Deep Successor Representation. ICLR 2018